# Impact of electrolyte abnormalities and adverse outcomes in persons with eating disorders: A systematic review protocol

Amos Buh [1]*, Mekaylah Scott[1,2], Rohan Kiska[1,3], Stephen G. Fung[4], Marco Solmi[1,5,6,7], Rachel Kang[1,2], Maria Salman [1,8], Kathryn Lee[1,9], Benjamin Milone[1,9], Gamal Wafy[1], Sarah Syed[1,8], Shan Dhaliwal[1,8], Maya Gibb[1], Ayub Akbari[1], Pierre A. Brown [1], Gregory L. Hundemer[1], Manish M. Sood[1,10]

1 Ottawa Hospital Research Institute, Ottawa, Ontario, Canada, 2 Faculty of Health Sciences, University of Ottawa, Ottawa, Ontario, Canada, 3 Faculty of Science, Carleton University, Ottawa, Ontario, Canada, 4 Bruyère Research Institute, Ottawa, Ontario, Canada, 5 Department of Psychiatry, University of Ottawa, Ottawa, Ontario, Canada, 6 Department of Mental Health, The Ottawa Hospital, Ottawa, Ontario, Canada, 7 Department of Child and Adolescent Psychiatry, Charité Universitätsmedizin Berlin, Berlin, Germany, 8 Faculty of Medicine, University of Ottawa, Ottawa, Ontario, Canada, 9 Faculty of Science, McMaster University, Hamilton, Ontario, Canada, 10 Department of Medicine, University of Ottawa at The Ottawa Hospital, Ottawa, Ontario, Canada

* abuh@ohri.ca

**Data Availability Statement:** No datasets were generated or analysed during the current study. All

## Abstract

### Background

Electrolytes (sodium, potassium, calcium, magnesium, chloride, phosphate) are required in specific amounts for proper functioning of the human body. Although the body has different organ systems, such as the kidneys, that regulate electrolyte levels in the blood, electrolyte abnormalities occur frequently in people with eating disorders. The objective of this review will be to examine the association between electrolyte imbalances and adverse outcomes in people with eating disorders.

### Methods

A systematic review of studies on eating and electrolyte disorders shall be conducted. Electronic searches shall be done in the Ovid MEDLINE, EMBASE, and PsycINFO databases. Selected studies shall include randomized control trials (RCTs), non-randomized controlled trials, and cross-sectional studies published in English or French. Quality appraisal of studies and a narrative synthesis of extracted data shall be conducted.

### Discussion

This review will synthesize existing evidence on electrolyte abnormalities in people with eating disorders. It will identify the type of electrolyte imbalances, their impact, and outcomes in people with eating disorders. We anticipate that information that will be useful to policy makers and clinicians in designing better policies to prevent eating disorders and or manage people with eating disorders shall be elucidated in this study.

relevant deidentified data from this study will be made available upon study completion.

**Funding:** The author(s) received no specific funding for this work.

**Competing interests:** The authors have declared that no competing interests exist.

**Abbreviations:** JBI, Joanna Briggs Institute; JBI-MASTARI, Joanna Briggs Institute Meta-Analysis of Statistics Assessment and Review instrument; PRISMA, Preferred Reporting Items for Systematic Reviews and Meta-Analyses; PROSPERO, International Prospective Register of Systematic Reviews; RCT, randomized Controlled Trial.

## Dissemination

The final manuscript will be submitted for publication in a journal.

## Review registration

This protocol has been registered with the International Prospective Register of Systematic Reviews (PROSPERO); registration number CRD42023477497.

## Background

Electrolytes are electrically charged molecules which play a number of vital roles in the normal functioning of the human body [1,2]. Major electrolytes found in the human body include potassium, sodium, calcium, magnesium, phosphate, chloride and bicarbonates [3,4]. Their extracellular concentrations in the body can be measured by blood tests which help in the assessment of a patient's clinical condition–an abnormally high or low level of electrolytes can disrupt body functions and can lead to life-threatening complications [2,3].

Evidence suggests that electrolyte imbalances (abnormally high or low levels of electrolytes falling outside the normal ranges or reference intervals) are associated with increased morbidity and mortality [5]. The causes are multifactorial and include physiological factors such as nutritional status, concurrent acid-base imbalances, pharmaceuticals, other co-morbid disorders or acute illnesses [1,5]. Besides these, refeeding syndrome (when someone who has been malnourished or starved begins feeding again) is a major causes electrolyte deficiencies [6–8]. The prevalence of electrolyte abnormalities have been reported to range between 15.0% to 44.1% [2,9,10]. The abnormalities are common among older community subjects and are mainly associated with diabetes mellitus and diuretics [9].

Nonetheless, despite advances in medicine, electrolyte abnormalities remain widespread and are also very commonly observed in persons with eating disorders [11]. Eating disorders such as bulimia nervosa and anorexia nervosa are disturbances in eating behaviors that compromises a person's health and impairs normal functioning [12]. Eating disorders often result in electrolyte abnormalities. Furthermore, some purging behaviors leading to electrolyte imbalances includes severe restriction of food, food binges, over exercising, and even activities such as forced vomiting, fasting, taking laxatives or diuretics, and performing enemas [10,11]. Some of the most common electrolyte imbalances caused by eating disorders include hypophosphatemia, hypokalemia, hyponatremia, and metabolic acidosis and alkalosis [11,13]. In fact, extracellular hypophosphatemia is the most commonly reported electrolyte disturbance resulting from a combination of cellular uptake of phosphorus together with depletion of total body stores during starvation [14–16].

The severity and type of electrolyte abnormalities are dependent on the mode, and frequency of the behaviors of the eating disorder [17]. Some signs and symptoms of electrolyte imbalances in people with eating disorders include constipation, nausea and vomiting, fatigue, heart palpitations or arrhythmias, muscle weakness, cramps, numbness, polyuria, headache, confusion, restlessness and irritability [18]. While complications of eating disorders range from minor to fatal, every organ system can be affected with problems ranging from pancytopenia to diffuse myalgias with muscle breakdown [19]. In fact, eating disorders can impact neurological, cardiac, muscle, gastrointestinal, endocrine and renal functions [11,18,20]. However, it has been documented that electrolyte abnormalities are associated with a host of serious

adverse clinical outcomes [12]. For instance, derangements in potassium, magnesium, and calcium levels predispose to cardiac arrhythmias and sudden cardiac death [21]. Derangements in sodium and phosphate levels can result in generalized weakness, cognitive impairment, and seizures [22,23]. Electrolyte disturbances can also have detrimental effects on kidney, gastrointestinal and bone health [24–26]. Nonetheless, there is little or no information on reviews that have comprehensively assessed the impact of electrolyte abnormalities and adverse outcomes resulting from people with eating disorders. The objective of this review therefore shall be to assess the association of electrolyte imbalances and adverse outcomes in people with eating disorders.

### Research question

What is the association of electrolyte abnormalities and adverse outcomes in people with eating disorders?

## Methods

### Study design

This shall be a systematic review of studies that have assessed electrolyte abnormalities in people with eating disorders. The review shall be conducted and reported following the Preferred Reporting Items for Systematic Reviews and Meta-Analyses Protocols (PRISMA-P) criteria (S1 File) [27]. A PRISMA diagram (S2 File) shall be used to show how studies shall be selected for this review. The review has been registered with the International Prospective Register of Systematic Reviews (PROSPERO)–number CRD42023477497.

### Inclusion criteria

**Population.**   The review shall include studies that involved people with eating disorders irrespective of age.

**Exposure.**   This review shall consider studies that evaluated patients with electrolyte abnormalities.

**Controls.**   The controls will be patients having no electrolyte abnormalities.

**Outcomes.**   This review will consider studies that shall include the following outcome measures:

- Proportion of hospitalization, worsening eating disorder / worsening mental health condition and relapse among patients resulting from their eating behavior.

- Proportion of people with delirium, ventricular tachycardia, QT interval prolongation, sudden cardiac arrest, refeeding syndrome, and death resulting from eating disorders related issues.

**Types of studies to be included.**   This review shall include experimental study designs including randomized control trials (RCTs), cohort and cross-sectional studies. Only studies conducted in English or French and involving people diagnosed with eating disorders shall be included in this review.

### Search strategy

This review will follow a three-step strategy to find studies conducted on eating and electrolyte disorders. Firstly, an initial search of the Ovid MEDLINE database using an analysis of text words found in the title and abstract, and the index terms used to describe the article shall be

conducted. Secondly, identified keywords and index terms in the first step shall be used to search for articles in other databases. Thirdly, the reference list of selected studies from the first and second steps shall be used to look for studies not found in the databases.

The databases that shall be searched for this review will include Ovid MEDLINE, EMBASE, and PsycINFO.

The initial keywords used for the searches in the Ovid MEDLINE database included *'Electrolyte disorder', 'eating disorder', 'dehydration', 'hypophosphatemia', 'hypokalemia', 'hypercalcemia', 'hyponatremia', 'metabolic acidosis', 'metabolic alkalosis', 'eating disorder hospitalisation', 'worsening eating disorder after hospitalisation', 'electrolyte disorder hospitalization'* (S3 File). These searches were conducted on March 26, 2024.

### Screening and selection process

All articles found in the searched databases will be imported into the Covidence software for screening. Two reviewers shall independently screen titles and abstracts to identify potentially relevant studies. Any disagreements shall be resolved through discussion. This same procedure shall be repeated in screening the full text of studies that shall be retained after the title and abstract screening.

### Assessment of methodological quality

Two independent reviewers shall assess the methodological validity of the studies that will be selected for retrieval prior to their inclusion in this review. The assessment shall be done using a standard critical appraisal tool from the Joanna Briggs Institute Meta-Analysis of Statistical Assessment and Review Instrument (JBI-MAStARI) (S4 File). Any disagreements between the two reviewers shall be settled through discussion.

### Data extraction

Data shall be extracted from selected studies independently by two reviewers, using a standardized data extraction tool from the Joanna Briggs Institute Meta-Analysis of Statistics Assessment and Review instrument (S5 File). The extracted data will include specific details about eating and electrolyte abnormalities, study populations, study methods and outcomes significant to the review question. In the event of any missing data from a study, the corresponding author of that study shall be contacted to provide the missing data.

### Data synthesis

We envision conducting both a meta-analysis and narrative synthesis if we have studies with information that can permit these analyses. The meta-analysis will be done to identify the eating disorders with a significant impact on development of electrolyte imbalances in an individual. For this analysis, we will first assess the statistical heterogeneity with $I^2$, which indicates the percentage of the total variation across studies; where 0% - 40% indicates low heterogeneity, 30% - 60% indicates moderate heterogeneity, 50% - 90% indicates substantial heterogeneity, and 75% - 100% indicates considerable heterogeneity. If there is a substantial amount of heterogeneity (75%), then sources of heterogeneity will be examined through subgroup and sensitivity analyses. We will also use Chi-square test to test the heterogeneity and consider P-values < 0.05 as statistically significant. We will select a fixed-effects model for significant homogeneous studies; otherwise, we will apply a random-effects model. We will summarize our outcomes using odds ratios (OR) and 95% confidence intervals (CI). We will consider an OR<1 to indicate a lower rate of outcome (electrolyte imbalance or abnormality) among the

group of participants who had a particular eating disorder. Publication bias will be assessed by visual inspections of funnel plots and Egger's test.

The narrative synthesis will involve a description of the eating disorders, electrolyte abnormalities and their association as extracted from selected studies. This synthesis shall be structured by describing studies according to type of eating disorders, electrolyte abnormalities and the outcome. The findings shall be presented in tables and figures where possible.

## Confidence in cumulative evidence

The quality of evidence in this review will be assessed by the Grades of Recommendation, Assessment, Development and Evaluation (GRADE)[28].

## Ethical considerations

No ethics clearance is required for this study as no primary data will be collected. The study will strictly adhere to the procedures outlined in this protocol in reviewing published and unpublished material on the review topic. However, in case of any amendments on this protocol, the amendments will be notified and registered. The final review manuscript will be published in a peer-reviewed journal at the end of this study.

## Discussion

This review will synthesize existing evidence on electrolyte abnormalities in people with eating disorders. We acknowledge that some reviews have been conducted on this area though with notable limitations including having varying objectives and being. For instance, a 2016 review was published on the approaches to refeeding in patients with anorexia nervosa. This review, besides being dated, focused exclusively on examining refeeding approaches in patients with anorexia nervosa and included articles published between 1960 and 2015 from the PubMed, PsycINFO, Scopus and Clinical Trials databases [29]. Similar to this, a 2023 review described the different nutritional interventions for children and adolescents with avoidant/restrictive food intake disorders [30]. A 2020 review on eating disorders during gestation described implications for mother's health, fetal outcomes, and epigenetic changes. This review included articles published between January 2000 and May 2020, but the review's focus was solely to clarify the mechanisms underpinning the adverse pregnancy outcomes in patients with eating disorders [31]. In yet another review investigating the relationship between weight and risk of medical instability in adolescents with typical and atypical anorexia nervosa, the specific medical instability risks assessed were limited to bradycardia, hypotension, hypothermia and hypophosphatemia and articles were included only from the EMBASE, Medline and PubMed databases [32]. In 2021, a nonstandard review reported common eating disorders in children and adolescent with an overview of treatment strategies [33], and a similar review in 2023 was conducted on renal and electrolyte complications in eating disorders [11]. Finally, a recent review published in 2024, only focused on providing a prevalence estimate of self-reported disordered eating and risk factors in athletes [34]. All of these prior reviews likely missed studies not published in English and those not found in the databases they assessed. More importantly, none of the reviews assessed the association between electrolyte abnormalities and adverse outcomes in people with eating disorders.

Given the limitations of the above published reviews, our review will identify the type of electrolyte imbalances, their impact, and associated outcomes in people with eating disorders.

### Strengths and limitations of this review

The major strengths of this review are that it will be a systematic review of randomized controlled trials and observational studies, and it will be the first to synthesize the evidence regarding the impact of electrolyte abnormalities on health outcomes in persons with eating disorders. Nonetheless, the main limitation of this review might be scarcity of randomized controlled trials and observational studies on impact of electrolyte abnormalities resulting from eating disorders, publication bias and methodological quality of the studies that shall be found. Also, there may be heterogeneity in eligible studies and incomplete information reported in included studies which could limit our ability to statistically assess the impact of electrolyte imbalances. However, we anticipate that the findings of this study will generate information that policy makers, clinicians, and stakeholders will find useful in designing better policies or strategies to prevent eating disorders and manage people with eating disorders or any related electrolyte abnormalities linked to eating disorders.

### Supporting information

**S1 File. PRISMA-P checklist.**
(DOCX)

**S2 File. PRISMA flow diagram.**
(PNG)

**S3 File. Medline search strategy.**
(PDF)

**S4 File. JBI assessment and review instrument.**
(PDF)

**S5 File. JBI data extraction form.**
(PDF)

### Acknowledgments

We acknowledge the librarian (Marie-Cécile Domecq) who guided us in developing the searching strategies.

### Author Contributions

**Conceptualization:** Amos Buh, Mekaylah Scott, Stephen G. Fung, Ayub Akbari, Pierre A. Brown, Gregory L. Hundemer, Manish M. Sood.

**Data curation:** Amos Buh, Mekaylah Scott, Stephen G. Fung, Ayub Akbari, Pierre A. Brown, Gregory L. Hundemer, Manish M. Sood.

**Formal analysis:** Amos Buh, Mekaylah Scott, Stephen G. Fung, Ayub Akbari, Pierre A. Brown, Gregory L. Hundemer, Manish M. Sood.

**Methodology:** Amos Buh, Mekaylah Scott, Rohan Kiska, Stephen G. Fung, Marco Solmi, Rachel Kang, Maria Salman, Kathryn Lee, Benjamin Milone, Gamal Wafy, Sarah Syed, Shan Dhaliwal, Maya Gibb, Ayub Akbari, Pierre A. Brown, Gregory L. Hundemer, Manish M. Sood.

**Supervision:** Ayub Akbari, Pierre A. Brown, Gregory L. Hundemer, Manish M. Sood.

**Validation:** Rohan Kiska, Marco Solmi, Rachel Kang, Maria Salman, Kathryn Lee, Benjamin Milone, Gamal Wafy, Sarah Syed, Shan Dhaliwal, Maya Gibb, Ayub Akbari, Pierre A. Brown, Gregory L. Hundemer, Manish M. Sood.

**Writing – original draft:** Amos Buh, Mekaylah Scott, Rohan Kiska, Stephen G. Fung, Marco Solmi, Rachel Kang, Maria Salman, Kathryn Lee, Benjamin Milone, Gamal Wafy, Sarah Syed, Shan Dhaliwal, Maya Gibb, Ayub Akbari, Pierre A. Brown, Gregory L. Hundemer, Manish M. Sood.

**Writing – review & editing:** Amos Buh, Mekaylah Scott, Rohan Kiska, Stephen G. Fung, Marco Solmi, Rachel Kang, Maria Salman, Kathryn Lee, Benjamin Milone, Gamal Wafy, Sarah Syed, Shan Dhaliwal, Maya Gibb, Ayub Akbari, Pierre A. Brown, Gregory L. Hundemer, Manish M. Sood.

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
