## [Decision Letter · Decision Letter 0]

10 Jun 2024

PONE-D-24-15566Impact of electrolyte abnormalities and adverse outcomes in persons with eating disorders: a systematic review protocol

PLOS ONE

Dear Dr. Buh,

Thank you for submitting your manuscript to PLOS ONE. After careful consideration, we feel that it has merit but does not fully meet PLOS ONE’s publication criteria as it currently stands. Therefore, we invite you to submit a revised version of the manuscript that addresses the points raised during the review process.

Please consider the following important points for enhancing the quality of your study protocol:

1. Importance of a Discussion Section:

The protocol currently lacks a discussion section. Including a discussion section is crucial as it allows the authors to interpret their findings in the context of existing literature, address potential implications of the study, and suggest directions for future research. Without this section, the readers are left without guidance on how to understand the significance and context of the study results.

2. Interpretation of Potential Outcomes:

The discussion section should include interpretations of potential outcomes. This involves hypothesizing possible results and explaining their significance. What do you expect to find, and what would these findings mean in the broader context of the field?

3. Integration with Existing Literature:

It is essential to discuss how your study fits within the existing body of literature. Compare and contrast your expected findings with previous studies and explain how your research could confirm, refute, or expand upon these studies.

4. Implications of the Study:

Elaborate on the potential implications of your study. How could your findings impact clinical practice, policy, or further research? Discuss both the theoretical and practical implications.

5. Addressing Unexpected Results:

Consider potential unexpected results if come and how they might be explained. Provide alternative hypothesis and what further research might be needed if the results are contrary to your hypotheses.

Comments on Missing Limitations Section:

1. Importance of Identifying Limitations:

The protocol does not include a limitations section. Acknowledging the limitations of your study is crucial for transparency and provides context for interpreting the results. It demonstrates a comprehensive understanding of your study design and the factors that could affect the validity and generalizability of the findings.

2. Methodological Limitations:

Detail any methodological limitations. For example, consider sample size, sampling methods, measurement tools, and potential biases. How might these factors limit the interpretation or generalizability of the results?

3. Generalizability of Findings:

Discuss any limitations related to the generalizability of your findings. For instance, if your sample is drawn from a specific population, explain how this might limit the applicability of the results to other groups.

4. Potential Confounding Factors:

Identify any potential confounding factors that could influence the results. How will you control for these, and what impact could they have if they are not adequately controlled?

5. Data Collection and Analysis Constraints:

Acknowledge any limitations related to data collection and analysis. For example, if there are potential issues with data accuracy, completeness, or the analytical methods used, these should be clearly stated.

General Recommendations:

1. Enhancing Transparency:

Including detailed discussion and limitations sections will enhance the the transparency and credibility of your study. It shows that you have thoroughly considered all aspects of your research design and are forthcoming about potential weaknesses.

2. Providing Comprehensive Insight:

These sections will provide readers with a comprehensive understanding of your study, helping them to better appreciate the significance, reliability, and applicability of your findings.

3. Strengthening the Study Protocol:

I strongly recommend adding these sections to strengthen the overall quality of your study protocol. A well-rounded protocol not only highlights the strengths of your study but also critically appraises its limitations and situates it within the larger research landscape. 

I hope these comments will guide the authors to improve their study protocol by ensuring it includes thorough discussion and limitations sections, thereby enhancing the overall quality and robustness of their research.

We look forward to receiving your revised manuscript.

Kind regards,

Muhammad Shahzad Aslam, Ph.D.,M.Phil., Pharm-D

Academic Editor

PLOS ONE

Additional Editor Comments:

Please consider the following important points for enhancing the quality of your study protocol:

1. Importance of a Discussion Section:

The protocol currently lacks a discussion section. Including a discussion section is crucial as it allows the authors to interpret their findings in the context of existing literature, address potential implications of the study, and suggest directions for future research. Without this section, the readers are left without guidance on how to understand the significance and context of the study results.

2. Interpretation of Potential Outcomes:

The discussion section should include interpretations of potential outcomes. This involves hypothesizing possible results and explaining their significance. What do you expect to find, and what would these findings mean in the broader context of the field?

3. Integration with Existing Literature:

It is essential to discuss how your study fits within the existing body of literature. Compare and contrast your expected findings with previous studies and explain how your research could confirm, refute, or expand upon these studies.

4. Implications of the Study:

Elaborate on the potential implications of your study. How could your findings impact clinical practice, policy, or further research? Discuss both the theoretical and practical implications.

5. Addressing Unexpected Results:

Consider potential unexpected results if come and how they might be explained. Provide alternative hypothesis and what further research might be needed if the results are contrary to your hypotheses.

Comments on Missing Limitations Section:

1. Importance of Identifying Limitations:

The protocol does not include a limitations section. Acknowledging the limitations of your study is crucial for transparency and provides context for interpreting the results. It demonstrates a comprehensive understanding of your study design and the factors that could affect the validity and generalizability of the findings.

2. Methodological Limitations:

Detail any methodological limitations. For example, consider sample size, sampling methods, measurement tools, and potential biases. How might these factors limit the interpretation or generalizability of the results?

3. Generalizability of Findings:

Discuss any limitations related to the generalizability of your findings. For instance, if your sample is drawn from a specific population, explain how this might limit the applicability of the results to other groups.

4. Potential Confounding Factors:

Identify any potential confounding factors that could influence the results. How will you control for these, and what impact could they have if they are not adequately controlled?

5. Data Collection and Analysis Constraints:

Acknowledge any limitations related to data collection and analysis. For example, if there are potential issues with data accuracy, completeness, or the analytical methods used, these should be clearly stated.

General Recommendations:

1. Enhancing Transparency:

Including detailed discussion and limitations sections will enhance the the transparency and credibility of your study. It shows that you have thoroughly considered all aspects of your research design and are forthcoming about potential weaknesses.

2. Providing Comprehensive Insight:

These sections will provide readers with a comprehensive understanding of your study, helping them to better appreciate the significance, reliability, and applicability of your findings.

3. Strengthening the Study Protocol:

I strongly recommend adding these sections to strengthen the overall quality of your study protocol. A well-rounded protocol not only highlights the strengths of your study but also critically appraises its limitations and situates it within the larger research landscape.

I hope these comments will guide the authors to improve their study protocol by ensuring it includes thorough discussion and limitations sections, thereby enhancing the overall quality and robustness of their research.

Reviewers' comments:

Reviewer's Responses to Questions

**Comments to the Author**

1. Does the manuscript provide a valid rationale for the proposed study, with clearly identified and justified research questions?

Reviewer #1: Yes

Reviewer #2: Yes

Reviewer #3: Yes

2. Is the protocol technically sound and planned in a manner that will lead to a meaningful outcome and allow testing the stated hypotheses?

Reviewer #1: Yes

Reviewer #2: Yes

Reviewer #3: Yes

3. Is the methodology feasible and described in sufficient detail to allow the work to be replicable?

Reviewer #1: Yes

Reviewer #2: Yes

Reviewer #3: Yes

4. Have the authors described where all data underlying the findings will be made available when the study is complete?

Reviewer #1: Yes

Reviewer #2: Yes

Reviewer #3: Yes

5. Is the manuscript presented in an intelligible fashion and written in standard English?

Reviewer #1: Yes

Reviewer #2: Yes

Reviewer #3: Yes

6. Review Comments to the Author

You may also provide optional suggestions and comments to authors that they might find helpful in planning their study.

Reviewer #1: In this study, authors aimed to assess the association of electrolyte imbalances and adverse outcomes in people with eating disorders. According to their research question, they explained their study designed in the methods section. However, I have not seen statistical analysis and discussion sections in the article.

Reviewer #2: Authors should check the supplementary file 1 which seems to be filled for a different study with the headline "Effective educational interventions for the promotion of sexual and reproductive health and rights for school-age children in low and middle-income countries: A systematic review protocol".

Reviewer #3: It would be really very interesting and novel review paper. Hope to open the new avenue of research in future on the same topic. Really exciting to see as soon in published form.

Categorically explain the aim, research question, background, study and methodology of the study. Upon on review the supplementary attached documents which more clear the novelty and comprehension of the stud.

However, minor changes in background chapter required to add more evidence-based studies on biological explanation how electrolytes imbalance in individual with eating disorder cause health adverse outcomes with sign and symptoms. Categorically explain the complications in ED as outcome of the electrolytes disorders.

7. PLOS authors have the option to publish the peer review history of their article (what does this mean?). If published, this will include your full peer review and any attached files.

Reviewer #1: No

Reviewer #2: No

Reviewer #3: **Yes: **Huma Naqeeb

---

## [Author Response · Author response to Decision Letter 0]

4 Jul 2024

July 04, 2024

Manuscript reference #: PONE-D-24-15566

Title: Impact of electrolyte abnormalities and adverse outcomes in persons with eating disorders: a systematic review protocol

RE: PLOS ONE Decision: Revision required [PONE-D-24-15566]

Dear Dr. Muhammad Shahzad Aslam,

We thank you, the editorial team, and the reviewers for these insightful and helpful comments and for giving us the chance to revise our manuscript. We believe the comments provided have further strengthened our manuscript and we are pleased to submit a revised version at this time. Please see below our point-by-point responses to the editorial and reviewer comments. 

Sincerely,

Amos Buh, PhD, FRSPH 

Corresponding Author 

Response to Reviewers’ reports

Response: We confirm that the manuscript meets PLOS ONE’s style requirements and all files have been named accordingly.

 Response: Thank you for this comment. In fact, all data that will be used in this study will be secondary data extracted from published papers. As such, the data will already be publicly available. Besides this, all data used in the study will be published together with the manuscript as tables or figures.

Additional Editor Comments:

Please consider the following important points for enhancing the quality of your study protocol:

1. Importance of a Discussion Section:

The protocol currently lacks a discussion section. Including a discussion section is crucial as it allows the authors to interpret their findings in the context of existing literature, address potential implications of the study, and suggest directions for future research. Without this section, the readers are left without guidance on how to understand the significance and context of the study results.

Response: We greatly appreciate this comment. A discussion section has now been added on the manuscript as requested (pages 8-9, lines 185-223): “Discussion

This review will synthesize existing evidence on electrolyte abnormalities in people with eating disorders. We acknowledge that some reviews have been conducted on this area though with notable limitations including having varying objectives and being. For instance, a 2016 review was published on the approaches to refeeding in patients with anorexia nervosa. This review, besides being dated, focused exclusively on examining refeeding approaches in patients with anorexia nervosa and included articles published between 1960 and 2015 from the PubMed, PsycINFO, Scopus and Clinical Trials databases [26]. Similar to this, a 2023 review described the different nutritional interventions for children and adolescents with avoidant/restrictive food intake disorders [27]. A 2020 review on eating disorders during gestation described implications for mother’s health, fetal outcomes, and epigenetic changes. This review included articles published between January 2000 and May 2020, but the review’s focus was solely to clarify the mechanisms underpinning the adverse pregnancy outcomes in patients with eating disorders [28]. In yet another review investigating the relationship between weight and risk of medical instability in adolescents with typical and atypical anorexia nervosa, the specific medical instability risks assessed were limited to bradycardia, hypotension, hypothermia and hypophosphatemia and articles were included only from the EMBASE, Medline and PubMed databases [29]. In 2021, a nonstandard review reported common eating disorders in children and adolescent with an overview of treatment strategies [30], and a similar review in 2023 was conducted on renal and electrolyte complications in eating disorders [11]. Finally, a recent review published in 2024, only focused on providing a prevalence estimate of self-reported disordered eating and risk factors in athletes [31]. All of these prior reviews likely missed studies not published in English and those not found in the databases they assessed. More importantly, none of the reviews assessed the association between electrolyte abnormalities and adverse outcomes in people with eating disorders.

Given the limitations of the above published reviews, our review will identify the type of electrolyte imbalances, their impact, and associated outcomes in people with eating disorders. 

Strengths and limitations of this review

The major strengths of this review are that it will be a systematic review of randomized controlled trials and observational studies, and it will be the first to synthesize the evidence regarding the impact of electrolyte abnormalities on health outcomes in persons with eating disorders. Nonetheless, the main limitation of this review might be scarcity of randomized controlled trials and observational studies on impact of electrolyte abnormalities resulting from eating disorders, publication bias and methodological quality of the studies that shall be found. Also, there may be heterogeneity in eligible studies and incomplete information reported in included studies which could limit our ability to statistically assess the impact of electrolyte imbalances. However, we anticipate that the findings of this study will generate information that policy makers, clinicians, and stakeholders will find useful in designing better policies or strategies to prevent eating disorders and manage people with eating disorders or any related electrolyte abnormalities linked to eating disorders.” 

2. Interpretation of Potential Outcomes:

The discussion section should include interpretations of potential outcomes. This involves hypothesizing possible results and explaining their significance. What do you expect to find, and what would these findings mean in the broader context of the field?

Response: This has been addressed in the discussion section as requested (please refer to the new text pasted above in response to Editorial comment #1).

3. Integration with Existing Literature:

It is essential to discuss how your study fits within the existing body of literature. Compare and contrast your expected findings with previous studies and explain how your research could confirm, refute, or expand upon these studies.

Response: Thank you for this suggestion; we have cited other studies on this topic in our discussion section and explained the limitations of the studies (please refer to the new text pasted above in response to Editorial comment #1). 

4. Implications of the Study:

Elaborate on the potential implications of your study. How could your findings impact clinical practice, policy, or further research? Discuss both the theoretical and practical implications.

Response: This has been addressed on the discussion section as requested. Please see page 9, lines 220-223 on the Revised Manuscript with Track Changes.

5. Addressing Unexpected Results:

Consider potential unexpected results if come and how they might be explained. Provide alternative hypothesis and what further research might be needed if the results are contrary to your hypotheses.

Response: Thank you for this suggestion. We intend to adhere strictly to the inclusion criteria setout in our protocol and data will be extracted only from studies that met the criteria and were included in the study. As such, it will be very unlikely to have unexpected results except there will be studies with missing information. Even in cases of missing information, we have explained in our protocol that the corresponding authors of studies with missing information will be contacted to provide such information. In the worst-case scenario where we are unable to reach a corresponding author, the given study will be excluded following our stated criteria. Given this, the possibility of having unexpected results is rolled out in our study – we strongly anticipate that our results will reflect the data extracted only from included studies that met our inclusion criteria.

Comments on Missing Limitations Section:

1. Importance of Identifying Limitations:

The protocol does not include a limitations section. Acknowledging the limitations of your study is crucial for transparency and provides context for interpreting the results. It demonstrates a comprehensive understanding of your study design and the factors that could affect the validity and generalizability of the findings.

Response: This section has been included as requested; please refer to the new text pasted above in response to Editorial comment #1.

2. Methodological Limitations:

Detail any methodological limitations. For example, consider sample size, sampling methods, measurement tools, and potential biases. How might these factors limit the interpretation or generalizability of the results?

Response: We greatly appreciate this comment; this has been addressed in the strengths and limitation section; page 9 lines 211-223 on the Revised Manuscript with Track Changes.

3. Generalizability of Findings:

Discuss any limitations related to the generalizability of your findings. For instance, if your sample is drawn from a specific population, explain how this might limit the applicability of the results to other groups.

Response: Thank you for this suggestion; any potential limitations of our study have been discussed under the strengths and limitation section of our protocol.

4. Potential Confounding Factors:

Identify any potential confounding factors that could influence the results. How will you control for these, and what impact could they have if they are not adequately controlled?

Response: Heterogeneity in included studies and differences in study participants characteristics across included studies could potentially influence our results. However, we have explained under data synthesis that we will conduct subgroup and sensitivity analysis if there is substantial heterogeneity among included studies.

5. Data Collection and Analysis Constraints:

Acknowledge any limitations related to data collection and analysis. For example, if there are potential issues with data accuracy, completeness, or the analytical methods used, these should be clearly stated.

Response: We appreciate this suggestion. The only issue we will face with data collection and analysis is missing data from included studies. However, we have described how this will be managed under data extraction and data synthesis on the protocol.

General Recommendations:

1. Enhancing Transparency:

Including detailed discussion and limitations sections will enhance the the transparency and credibility of your study. It shows that you have thoroughly considered all aspects of your research design and are forthcoming about potential weaknesses.

Response: Thank you. As recommended, these sections have been included on the manuscript. Please refer to the new text pasted above in response to Editorial comment #1. 

2. Providing Comprehensive Insight:

These sections will provide readers with a comprehensive understanding of your study, helping them to better appreciate the significance, reliability, and applicability of your findings.

Response: We appreciate this greatly and the requested sections have been included on the manuscript (please refer to the new text pasted above in response to Editorial comment #1).Thank you.

3. Strengthening the Study Protocol:

I strongly recommend adding these sections to strengthen the overall quality of your study protocol. A well-rounded protocol not only highlights the strengths of your study but also critically appraises its limitations and situates it within the larger research landscape.

Response: Thank you for this recommendation. We have added the recommended sections and we believe the manuscript has been greatly improved.

I hope these comments will guide the authors to improve their study protocol by ensuring it includes thorough discussion and limitations sections, thereby enhancing the overall quality and robustness of their research.

Response: Indeed, the comments were very constructive and have helped us improve the quality of our manuscript. We are very grateful to the editor and all reviewers; thank you.

Reviewers' comments:

Reviewer's Responses to Questions

Comments to the Author

1. Does the manuscript provide a valid rationale for the proposed study, with clearly identified and justified research questions?

Reviewer #1: Yes

Reviewer #2: Yes

Reviewer #3: Yes

Response: Thank you so much for your appraisal of our manuscript. 

2. Is the protocol technically sound and planned in a manner that will lead to a meaningful outcome and allow testing the stated hypotheses?

Reviewer #1: Yes

Reviewer #2: Yes

Reviewer #3: Yes

Response: We greatly appreciate your appraisal of our manuscript, thank you.

3. Is the methodology feasible and described in sufficient detail to allow the work to be replicable?

Reviewer #1: Yes

Reviewer #2: Yes

Reviewer #3: Yes

Response: We greatly appreciate your appraisal of our manuscript, thank you. 

4. Have the authors described where all data underlying the findings will be made available when the study is complete?

Reviewer #1: Yes

Reviewer #2: Yes

Reviewer #3: Yes

Response: Thank you.

5. Is the manuscript presented in an intelligible fashion and written in standard English?

PLOS ONE does not copyedit accepted 

---

## [Editor Report · Decision Letter 1]

9 Jul 2024

PONE-D-24-15566R1Impact of electrolyte abnormalities and adverse outcomes in persons with eating disorders: a systematic review protocolPLOS ONE

Dear Dr. Buh,

Thank you for submitting your manuscript to PLOS ONE. After careful consideration, we feel that it has merit but does not fully meet PLOS ONE’s publication criteria as it currently stands. Therefore, we invite you to submit a revised version of the manuscript that addresses the points raised during the review process.

Authors should check the supplementary file 1 which seems to be filled for a different study with the headline "Effective educational interventions for the promotion of sexual and reproductive health and rights for school-age children in low and middle-income countries: A systematic review protocol".

It would be really very interesting and novel review paper. Hope to open the new avenue of research in future on the same topic. Really exciting to see as soon in published form.

Categorically explain the aim, research question, background, study and methodology of the study. Upon on review the supplementary attached documents which more clear the novelty and comprehension of the stud.

However, minor changes in background chapter required to add more evidence-based studies on biological explanation how electrolytes imbalance in individual with eating disorder cause health adverse outcomes with sign and symptoms. Categorically explain the complications in ED as outcome of the electrolytes disorders.

We look forward to receiving your revised manuscript.

Kind regards,

Muhammad Shahzad Aslam, Ph.D.,M.Phil., Pharm-D

Academic Editor

PLOS ONE
---

## [Author Response · Author response to Decision Letter 1]

14 Jul 2024

July 13, 2024

Manuscript reference #: PONE-D-24-15566R1

Title: Impact of electrolyte abnormalities and adverse outcomes in persons with eating disorders: a systematic review protocol

RE: PLOS ONE Decision: Revision required [PONE-D-24-15566R1] - [EMID:4c1da07e9e1ed7f8]

Dear Dr. Muhammad Shahzad Aslam,

We are very grateful for the comments and for the opportunity to revise our manuscript further. We believe we have addressed all comments and are happy to submit a revised version of the manuscript. Please see below a point-by-point response to the comments. 

Sincerely,

Amos Buh, PhD, FRSPH 

Corresponding Author 

Response to Reviewers’ reports

Authors should check the supplementary file 1 which seems to be filled for a different study with the headline "Effective educational interventions for the promotion of sexual and reproductive health and rights for school-age children in low and middle-income countries: A systematic review protocol".

Response: We greatly appreciate this observation. We made an error with the headline while filling this file; the error has been corrected and the headline now reads “Impact of electrolyte abnormalities and adverse outcomes in persons with eating disorders: a systematic review protocol”. Thank you.

It would be really very interesting and novel review paper. Hope to open the new avenue of research in future on the same topic. Really exciting to see as soon in published form.

Response: Thank you so much for appraising our manuscript. We hope our work on this topic will inspire future research on this same topic. Once we complete the review, we will submit the final manuscript for publication so that everyone can have access to our findings.

Categorically explain the aim, research question, background, study and methodology of the study. Upon on review the supplementary attached documents which more clear the novelty and comprehension of the stud.

Response: We are deeply grateful to you for taking a keen look on our manuscript and appreciating its clarity from the aim through the research question, background and study methods to the supplementary documents. Thank you.

However, minor changes in background chapter required to add more evidence-based studies on biological explanation how electrolytes imbalance in individual with eating disorder cause health adverse outcomes with sign and symptoms. Categorically explain the complications in ED as outcome of the electrolytes disorders.

Response: Thank you for this suggestion. We have expanded our background section (particularly the last paragraph of the background) to include additional text regarding the existing literature on how electrolyte disorders may lead to adverse health outcomes (pages 3-4, lines 77-93): “The severity and type of electrolyte abnormalities are dependent on the mode, and frequency of the behaviors of the eating disorder [17]. Some signs and symptoms of electrolyte imbalances in people with eating disorders include constipation, nausea and vomiting, fatigue, heart palpitations or arrhythmias, muscle weakness, cramps, numbness, polyuria, headache, confusion, restlessness and irritability [18]. While complications of eating disorders range from minor to fatal, every organ system can be affected with problems ranging from pancytopenia to diffuse myalgias with muscle breakdown [19]. In fact, eating disorders can impact neurological, cardiac, muscle, gastrointestinal, endocrine and renal functions [11, 18, 20]. However, it has been documented that electrolyte abnormalities are associated with a host of serious adverse clinical outcomes [12]. For instance, derangements in potassium, magnesium, and calcium levels predispose to cardiac arrhythmias and sudden cardiac death [18]. Derangements in sodium and phosphate levels can result in generalized weakness, cognitive impairment, and seizures [19, 20]. Electrolyte disturbances can also have detrimental effects on kidney, gastrointestinal and bone health [21 - 23]. Nonetheless, there is little or no information on reviews that have comprehensively assessed the impact of electrolyte abnormalities and adverse outcomes resulting from people with eating disorders.”

---

## [Editor Report · Decision Letter 2]

16 Jul 2024

Impact of electrolyte abnormalities and adverse outcomes in persons with eating disorders: a systematic review protocol

PONE-D-24-15566R2

Dear Dr. Buh,

We’re pleased to inform you that your manuscript has been judged scientifically suitable for publication and will be formally accepted for publication once it meets all outstanding technical requirements.

Kind regards,

Muhammad Shahzad Aslam, Ph.D.,M.Phil., Pharm-D

Academic Editor

PLOS ONE
---

## [Editor Report · Acceptance letter]

29 Jul 2024

PONE-D-24-15566R2 

PLOS ONE

Dear Dr. Buh, 

I'm pleased to inform you that your manuscript has been deemed suitable for publication in PLOS ONE. Congratulations! Your manuscript is now being handed over to our production team.

Kind regards, 

on behalf of

Dr. Muhammad Shahzad Aslam 

Academic Editor

PLOS ONE